# Investigating the Role of Circulating miRNAs as Biomarkers in Colorectal Cancer: An Epidemiological Systematic Review

**DOI:** 10.3390/biomedicines10092224

**Published:** 2022-09-08

**Authors:** Lucia Dansero, Fulvio Ricceri, Laura De Marco, Valentina Fiano, Ginevra Nesi, Lisa Padroni, Lorenzo Milani, Saverio Caini, Giovanna Masala, Claudia Agnoli, Carlotta Sacerdote

**Affiliations:** 1Department of Clinical and Biological Sciences, University of Turin, 10043 Turin, Italy; 2Unit of Cancer Epidemiology, Città della Salute e della Scienza University-Hospital and Department of Medical Sciences, University of Turin, 10126 Turin, Italy; 3Institute for Cancer Research, Prevention Clinical Network (ISPRO), 50139 Florence, Italy; 4Epidemiology and Prevention Unit, Fondazione IRCCS Istituto Nazionale dei Tumori, 20133 Milan, Italy

**Keywords:** microRNA, colorectal cancer, systematic review, biomarker

## Abstract

Colorectal cancer (CRC) is one of the most common cancers worldwide. Primary and secondary preventions are key to reducing the global burden. MicroRNAs (miRNAs) are a group of small non-coding RNA molecules, which seem to have a role either as tumor suppressor genes or oncogenes and to be related to cancer risk factors, such as obesity and inflammation. We conducted a systematic review and meta-analysis to identify circulating miRNAs related to CRC diagnosis that could be selected as biomarkers in a meet-in-the-middle analysis. Forty-four studies were included in the systematic review and nine studies in the meta-analysis. The pooled sensitivity and specificity of miR-21 for CRC diagnosis were 77% (95% CI: 69–84) and 82% (95% CI: 70–90), respectively, with an AUC of 0.86 (95% CI: 0.82–0.88). Several miRNAs were found to be dysregulated, distinguishing patients with CRC from healthy controls. However, little consistency was present across the included studies, making it challenging to identify specific miRNAs, which were consistently validated. Understanding the mechanisms by which miRNAs become biologically embedded in cancer initiation and promotion may help better understand cancer pathways to develop more effective prevention strategies and therapy approaches.

## 1. Introduction

Colorectal cancer (CRC) is one of the most common cancers worldwide, ranking third in terms of incidence (with more than 1.9 million new cases estimated in 2020) and second in mortality (with 935,000 deaths estimated in 2020) [1]. A substantial proportion of cases and deaths are attributable to modifiable and preventable risk factors. Several studies investigated these preventable risk factors for colorectal cancer, indicating sedentary lifestyle, excess body weight, heavy alcohol consumption, smoking, unhealthy diet, and physical inactivity [1,2,3]. Moreover, secondary prevention through screening is pivotal to reduce the rising global burden of CRC [4].

MicroRNAs (miRNAs) are a group of small non-coding RNA molecules that post-transcriptionally regulate gene expression by mRNA cleavage, mRNA destabilization, or inhibition of translation [5]. MiRNAs are extensively implicated in many complex physiological processes, such as cell proliferation, metabolism, and signal transduction [6]; further, it is not clear if they have a causal role in tumorigenesis or if their changes in tissues and blood are a consequence of cancer [7].

Several studies investigated the role of miRNAs as biomarkers for cancer detection and measured the values of area under the curve (AUC), sensitivity, and specificity in relation to dysregulation in a single miRNA or in a panel of miRNAs [8]. Many of them reported promising results; however, some dysregulations seem to be common in most cancers, possibly due to their role in cancer-associated biological processes and not in aetiology targets [8].

The aetiological association between miRNA dysregulation and CRC could be explained by several physiological processes, also related indirectly to CRC risk factors (such as obesity and inflammation) that could be affected by these molecules. For instance, a few studies investigated the association between obesity and circulating miRNA profile, detecting a number of these molecules as over-expressed in obese subjects [9]. Furthermore, a pattern of circulating miRNA expression linked to low-grade inflammation is being increasingly recognized, opening new perspectives in the study of intermediate biomarkers for cancer [10].

In this study, we aimed to review the relevant existing literature to identify published studies reporting the use of plasma, serum, or blood-based circulating miRNAs as biomarkers for the diagnosis of CRC. Identifying a preclinical biomarker of disease, such as miRNAs or a panel of miRNAs, that overlaps a marker of exposure (such as obesity or inflammation), would strengthen causal links between these exposures and the disease in a meet-in-the-middle approach. 

## 2. Materials and Methods

A systematic review was conducted following the PRISMA 2020 reporting guidelines [11] (see Appendix A). The study protocol was registered in the International Platform of Registered Systematic Review and Meta-analysis Protocols (INPLASY; registration number: 202290002).

PubMed and EMBASE were searched to identify published studies between the 1st of January 2012 (to exclude the few previous articles that used non-standardized laboratory procedures) and the 1st of April 2022 (end date of our search). The search keywords included “miRNA or microRNA”, “colon or colorectal cancer”, “circulating or exosomal” with restrictions to “humans”, excluding “cells”, and “tissue” (Appendix A details the full search strategy). Additional references were identified through citation searches and screening of relevant reviews. 

After removing duplicates, titles and abstracts were screened for any mention of miRNAs as biomarkers and colorectal cancer by two independent reviewers (LD and CS). In the presence of disagreement among the two reviewers, a third reviewer (FR) was consulted to settle the controversy.

### 2.1. Inclusion and Exclusion Criteria

Studies were considered eligible for the systematic review if they met the following criteria: (1) study patients were diagnosed with CRC; (2) healthy individuals were used as controls; (3) biological samples were plasma or serum or blood; (4) results included any measure of AUC, sensitivity, and specificity and/or fold change values.

Studies were excluded if they were: (1) reviews, meta-analyses, conference abstracts, or letters; (2) animal or cell experiments; (3) studies that investigated prognosis, survival, or metastatic cancers only; (4) studies that investigated toxicity or therapy efficacy; (5) studies with insufficient data; (6) studies not published in English language. 

We proceeded with the meta-analysis when at least three studies with sufficient information were found for the same miRNA. Studies in which the frequencies of true positives (TP), true negatives (TN), false positives (FP), and false negatives (FN) were not available or could not be calculated were excluded. 

### 2.2. Data Extraction

Once full texts were selected, references were screened to search for other relevant studies. According to inclusion criteria, data were extracted by two independent authors. Disagreements were solved through the involvement of a third author. 

Extracted data form included: first author’s name and reference, country, sample size, biological sample (plasma, serum, or blood), miRNA, cut-off value, AUC value (95% CI), sensitivity (95% CI), specificity (95% CI), fold change (95% CI), *p*-value, median relative expression (s.d.), miRNA source (candidate or discovery if found in a screening phase), and expression (up- or down-regulation). 

Diagnostic performance data were extracted or calculated for the studies included in the meta-analysis (FP, FN, TP, TN). 

### 2.3. Quality Assessment

Included studies were evaluated according to the Quality Assessment of Diagnostic Accuracy Studies 2 (QUADAS-2) checklist [12] to assess the risk of bias and applicability of studies of diagnostic accuracy (Appendix A). 

### 2.4. Statistical Analysis

STATA17 was used for the statistical analysis. A contingency table was calculated for each study included in the meta-analysis. A random effect model was applied to calculate sensitivity, specificity, positive likelihood ratio (PLR), negative likelihood ratio (NLR), and diagnostic odds ratio (DOR). The summary receiver operating characteristic curve (SROC) was then generated, together with the pooled sensitivity and specificity. Heterogeneity between studies was evaluated using the I^2^ statistics by Zhou and Dendukuri. Publication bias was not evaluated because of the small size of studies.

## 3. Results

### 3.1. Systematic Review

First, 369 records from the database search and 6 from manual search were identified (Figure 1); 285 records remained after duplicate removal; a further 219 records were excluded after the title and abstract reading as they were not relevant to the review topic (they were either reviews, meta-analyses, conference abstracts, animal or cell studies, or focused on other cancers). 

Following this, 66 full texts from the database search and 6 from the manual search were screened; 28 articles were subsequently excluded for the following reasons: review or meta-analysis article (1), not CRC (5), no available data (5), study on prognosis or survival (8), abstract only (1), inability to obtain full article (6), no healthy controls (1), and no plasma, serum, or blood sample (1). Finally, 44 studies that met the inclusion criteria were included in the systematic review. 

Table 1 describes all the study characteristics and outcomes of the studies included in the review.

**Table 1 biomedicines-10-02224-t001:** Study characteristics of the forty-four studies included in the systematic review.

First Author, Year	Country	Cases *(Mean Age + SD/ Median + Range)*	Controls *(Mean Age + SD/* *Median + Range)*	Biological Sample	miRNA	miRNA Source	Expression
Silva, 2021 [13]	Brazil	41 -	68 -	plasma	miR-106a-5p	discovery	up
					miR-542-5p	discovery	up
					let-7e-5p	discovery	up
					miR-28-3p	discovery	up
Han, 2021 [14]	China	123 *(51.60 ± 11.4)*	150 *(52.30 ± 11.25)*	serum	miR-15b	candidate	up
					miR-16	candidate	up
					miR-21	candidate	up
					miR-31	candidate	up
					Panel miRNA-15, miRNA-21, MiRNA-32	candidate	
Peng, 2020 [15]	China	80 *(61.08 ± 12.69)*	88 *(60.93 ± 10.89)*	serum	miR-30e-3p	discovery	up
					miR-31-5p	discovery	up
					miR-34b-3p	discovery	up
					miR-146a-5p	discovery	up
					miR-148a-3p	discovery	down
					miR-192-5p	discovery	down
Liu, 2020 [16]	China	80 -	23 -	plasma	miRNA-139-3p	candidate	down
Pastor-navarro, 2020 [17]	Spain	27 *(70*; *43–86)*	45 *(56.5*; *50–73)*	serum	miR-21	candidate	up
Li, 2020 [18]	China	597 *(62.89)*	585 *(57.17)*	plasma	miR-20b-5p	discovery	up
					miR-329-3p	discovery	up
					miR-374b-5p	discovery	up
					miR-503-5p	discovery	up
Bader El Din, 2020 [19]	Egypt	60 *(45.9 ± 9.8)*	30 *(42.1 ± 10.8)*	serum	let-7c	discovery	up
					miR-21	discovery	up
					miR-26a	discovery	up
					miR-146a	discovery	up
Zhao, 2019 [20]	China	169 -	155 -	serum	mir-150-5p	discovery	down
					miR-99b-5p	discovery	down
Sabry, 2018 [21]	Egypt	35 *(48.47 ± 15.16)*	101 *(49.59 ± 13.99)*	blood	miR-210	candidate	up
					miR-21	candidate	up
					miR-126	candidate	down
Marcuello, 2019 [22]	Spain	59 *(62.05)*	80 *(62.02)*	serum	miR-29a-3p, miR-15b-5p, miR-18a-5p, miR-19a-3p, miR-19b-3p, miR-335-5p	candidate	-
Liu, 2019 [23]	China	80 *(63.7 ± 9.2)*	30 *(59.4 ± 10.3)*	plasma	miR-1290	discovery	up
					miR-320d	discovery	down
Karimi, 2018 [24]	Iran	25 *(58.7)*	13	serum	miR-301a	discovery	up
					miR-23a	discovery	up
Villanueva, 2018 [25]	Spain	96 *(72.0)*	100 *(60.3)*	plasma	(miRNA19a, miRNA19b, miRNA15b, miRNA29a, miRNA335, miRNA18a)	candidate	-
Yang, 2018 [26]	China	46 *(60.67 ± 12.49)*	33 *(42.39 ± 13.13)*	serum	miR-20a	candidate	down
					miR-486	candidate	down
Liu, 2018 [27]	China	40 *(52.8 ± 6.2)*	40 *(51.4 ± 5.8)*	plasma	miR-27a	discovery	up
					miR-130a	discovery	up
Bilegsaikhan, 2018 [28]	China	80 *(59.2 ± 11.1)*	80 *(60.6 ± 11.3)*	serum	miR-338-5p	candidate	up
Wikberg, 2018 [29]	Sweden	67 -	134 -	plasma	miR-21	candidate	up
					miR-18a	candidate	up
					miR-25	candidate	up
					miR-22	candidate	up
Yan, 2017 [30]	China	77 -	20 -	serum	miR-18a	candidate	up
					miR-25	candidate	up
					miR-22	candidate	up
					miR-6869-5p	discovery	down
					miR-548c-5p	discovery	down
					miR-486-5p	discovery	up
					miR-3180-5p	discovery	up
Wang, 2017 [31]	China	50 -	44 -	serum	miR-31	discovery	up
					miR-141	discovery	up
					miR-224-3p	discovery	up
					miR-576-5p	discovery	up
					miR-4669	discovery	down
Wang, 2016 [32]	China	268 *(58; 49–66)*	102 *(56; 48–66)*	serum	miR-210	candidate	up
Wang, 2017 [33]	China	50 *(62.3)*	50 -	plasma	miR-125a-3p	discovery	up
					miR-320c	discovery	up
Ng, 2017 [34]	Hong Kong	117 -	90 -	serum	miR-139-3p	candidate	up
Zhang, 2017 [35]	China	20 -	20 -	serum	miR-4463	discovery	up
					miR-5704	discovery	up
					miR-371b-3p	discovery	down
					miR-1247-5p	discovery	down
					miR-1293	discovery	down
					miR-548at-5p	discovery	down
					miR-107	discovery	down
					miR-139-3p	discovery	down
Pan, 2017 [36]	China	80 *(63.75 ± 12.34)*	80 *(62.25 ± 8.24)*	plasma	miR-15b	discovery	up
					miR-17	discovery	up
					miR-21	discovery	up
					miR-26b	discovery	up
					miR-145	discovery	up
Bastaminejad, 2017 [37]	Iran	40 -	40 -	serum	miR-21	candidate	up
Sazanov, 2016 [38]	Russia	31 -	34 -	plasma	miR-21	candidate	up
Zekri, 2016 [39]	Egypt	100 *(46.7 ± 14.5)*	24 *(43.7 ± 14.2)*	serum	miR-223	candidate	up
					miR-17	candidate	up
					miR-19a	candidate	up
					miR-20a	candidate	up
Vychytilova-Faltejskova, 2016 [40]	Czech Republic	203 -	100 -	serum	miR-142-5p	discovery	up
					miR-23a-3p	discovery	up
					miR-27a-3p	discovery	up
					miR-376c-3p	discovery	up
Chen, 2016 [41]	USA	31 *(63.71)*	52 *(59.06)*	plasma	miR-21	candidate	up
					miR-152	candidate	up
Sarlinova, 2016 [42]	Slovakia	71 -	80 -	blood	miR-21	candidate	up
					miR-221	candidate	up
					miR-150	candidate	down
Basati, 2015 [43]	Iran	55 *(58.52 ± 10.02)*	55 *(57.87 ± 10.15)*	serum	miR-194	candidate	down
					miR-29b	candidate	down
Yamada, 2015 [44]	Japan	136 *(68)*	52 *(58)*	serum	miR-21	discovery	up
					miR-29a	discovery	up
					miR-125b	discovery	up
Nonaka, 2015 [45]	Japan	84 -	32 -	serum	miR-103	discovery	up
					miR-720	discovery	up
					miR-21	candidate	up
Ghanbari, 2014 [46]	Iran	61 *(64.13 ± 8.673)*	24 *(61.96 ± 8.67)*	plasma	miR-142-3p	discovery	down
					miR-26a-5p	discovery	down
Fang, 2015 [47]	China	111 *(60)*	43 -	plasma	miR-24	candidate	down
					miR-320a	candidate	down
					miR-423-5p	candidate	down
Chen, 2015 [48]	China	100 -	79 -	plasma	miR-106a	candidate	up
					miR-20a	candidate	up
Li, 2015 [49]	China	200 *(66.3 + 11.8)*	400 *(65.5 + 10.8)*	plasma	miR-29b	candidate	down
Wang, 2014 [50]	China	83 *(57 ± 10.4)*	59 *(55 ± 7.6)*	serum	miR-21	discovery	up
					Let-7g	discovery	up
					miR-31	discovery	down
					miR-92a	discovery	down
					miR-181b	discovery	down
					miR-203	discovery	down
Nonaka, 2014 [51]	Japan	114 -	32 -	serum	miR-199a-3p	discovery	up
					miR-21	candidate	up
Basati, 2014 [52]	Iran	40 *(55.35 ± 10.13)*	40 *(55.00 ± 10.35)*	serum	miR-21	Candidate	up
Giraldez, 2013 [53]	Spain	53 -	42 -	plasma	miR-19b	discovery	up
					miR-15b	discovery	up
					miR-29a	discovery	up
					miR-335	discovery	up
Luo, 2013 [54]	Germany	80 *(68.0± 10.4)*	144 *(62.5 ± 7.5)*	plasma	Panel: miR-29a, -106b, -133a, -342-3p, -532-3p-miR-18a, -20a, -21, -92a, -143, -145, -181b	Discovery and candidate	
Wang, 2012 [55]	China	90 *(62 ± 11)*	58 *(58 ± 12**)*	plasma	miR-601	discovery	down
					miR-760	discovery	down
Wang, 2012 [56]	China	32 *(63; 45–80)*	39 -	serum	miR-21	candidate	up

Overall, twenty-one were in China, five in Iran, four in Spain, three in Egypt, three in Japan, and one each in Brazil, USA, Czech Republic, Sweden, Germany, Russia, Slovakia, and Hong Kong (see Table 1). 

Twenty-one studies performed a miRNA screening phase to identify the dysregulated miRNAs in the plasma or serum of CRC patients and healthy controls to select the miRNAs for the validation phase, whereas the remaining studies selected the miRNAs based on the literature review or studies that they had previously conducted. 

Regarding the biological sample type, only 2 studies reported the use of blood samples, whereas 24 studies used serum samples and 18 plasma samples (Table 1). 

Overall, AUC values were the most used outcomes in the included studies (Appendix A); only 19 studies reported fold-change values and corresponding *p*-values (Appendix A). It was possible to extract data on relative expression for nine studies only, as many studies used graphical presentations of data without reporting the exact value (Appendix A). 

Overall, only eight studies reported all the selected outcomes (AUC values, sensitivity, specificity, fold change, and *p*-value) [19,20,21,36,42,43,53,56].

The miRNAs that were not validated by qRT-PCR or other methods were not included in this review. 

Overall, many dysregulated miRNAs were found, but most of them were not consistently analysed in more than one study. MiR-21 was the most-reported microRNA over the included studies, with 16 studies ([14,17,19,21,29,36,37,38,41,44,45,50,51,55], followed by miR-31([14,31,50]), miR-15b ([14,36,53]), and miR-20a ([15,39,48]), which were used in three studies each. However, in these studies, the expression of miR-20a did not have coherent values. Eight other miRNAs were analysed in two studies each (miR-210, miR-25, miR-139-3p, miR-29b, miR-18a, miR-22, miR-17, and miR-29a). These figures do not take into account the miRNAs used in panel. Homogenous results were found for miRNAs that were analysed in more than one study with no disagreement in terms of miRNA expression (up or down-regulation), except for miR-20a, which did not have consistent results.

The majority of the included studies analysed the expression of single miRNAs; however, nine studies included a panel of miRNAs (ranging from two to twelve miRNAs in each panel). 

### 3.2. Quality Assessment

Quality assessment using the QUADAS-2 tool of the 44 included studies is shown in Figure 2. Overall, the included studies had a low risk of bias and applicability regarding the reference standard and the flow and timing categories. However, unclear risk of bias was found in many studies for the patient selection and the index test. Indeed, many of them did not provide enough detail on the patient selection process or seemed to choose a case-control design. Regarding the index test category, some of the studies did not mention whether the threshold was prespecified. 

### 3.3. Meta-Analysis

Nine studies regarding one miRNA (miR-21) met the inclusion criteria and were included in the meta-analysis. All the studies showed an over-expression of circulating miR-21 in colon cancer cases, measured in serum, plasma, or whole blood. The pooled sensitivity and specificity of circulating miR-21 for CRC diagnosis were 77% (95% CI: 69–84) and 82% (95% CU: 70–90), respectively. Higher heterogeneity was found in specificity than sensitivity (specificity: *σ*^2^ = 0.85, I^2^ = 80.15; sensitivity: *σ*^2^ = 0.33, I^2^ = 72.05). Figure 3 shows a forest plot for the included studies. 

Diagnostic odds ratio for this meta-analysis was 15.34 (95% CU: 6.16–38.18), while PLR and NLR were 4.25 (95% CU: 2.35–7.68) and 0.28 (95% CU: 0.19–0.41), respectively. 

The ROC showed good diagnostic accuracy, with an AUC of 0.86 (95% CI: 0.82–0.88) for the included studies (Figure 4). 

Following the Cochrane guidelines, we did not perform publication bias analyses since the sample of included studies was too small [57]. However, it is likely that a publication bias exists, which could be due to the tendency of authors to report on articles only top-performing diagnostic miRNAs instead of all the tested miRNAs. Furthermore, in literature full of small-sample studies, models with worse diagnostic performance, mainly reflected in specificity, have a lower probability to be published. 

We performed meta-regression to investigate the possible source of heterogeneity. Biological sample type (plasma vs serum), reference miRNA used for normalization, and method for choosing the studied miRNA (literature vs. discovery) were used as covariates in the meta-regression. Statistically significant heterogeneity was found for the sample type, indicating that studies using plasma samples had a smaller heterogeneity than studies using serum samples. However, both groups had a small sample size, thus we could not conduct further analysis. 

## 4. Discussion

This systematic review of 44 articles investigating the role of plasma, serum, or blood-based miRNAs as biomarkers for CRC identified several miRNAs reported to be dysregulated.

We identified thirteen different studies that reported the up-regulation of miR-21; this scenario was expected, as miR-21 is one of the most investigated miRNAs in the literature, both for CRC and other cancers. In addition to miR-21, other miRNAs have been reported in more than one study, such as miR-31 (see Table 1); however, most of the miRNAs found in this systematic review were not validated more than once. This makes the comparison between studies challenging. 

Overall, little consistency was observed across the included studies, not only in terms of miRNA selection and validation, but also regarding outcome reporting, study design, sample type, normalization techniques, and patient selection. Above all, a major inconsistency is related to the selection of miRNAs; only around half of the studies performed discovery phase screening for miRNA expression in CRC patients and healthy controls, whereas the others selected the miRNAs from the literature. Moreover, among the studies that had a discovery phase, there is still inconsistency in the miRNAs that were found to be dysregulated. This is likely due to the choice of biological specimen type, the lack of standardization in laboratory techniques, sample size, and statistical analysis. Furthermore, data normalization for miRNA measurement is still an important challenge, which could lead to variability in the results [58].

Overall, using the QUADAS-2 tool, we found a high percentage of studies with low quality on patient selection and index test categories. Many studies did not provide enough detail on the patient selection methods or used a design similar to a case-control study. In most studies, the sample size was relatively small, especially for the discovery phase, and only a few studies reported an adequate inclusion criterion for control selection and matching. 

Only the study by Wikberg et al. [29] used a cohort study design, using pre- and post-diagnostic samples; all the others use a case-control design. 

Furthermore, more than half of the studies were conducted in one country (China), which did not allow us to consider genetic heterogeneity. 

The lack of consistency is one of the main limitations in the existing literature on miRNAs and cancer. Indeed, the strong heterogeneity in the studies included in terms of study design, miRNA selection, and outcomes and analysis could influence the validity of the results, especially for the miRNAs that were validated in one study only. 

In this study, we conducted a meta-analysis on miR-21 only, as the other miRNAs were not investigated more than two or three times. This did not allow us to conduct a meta-analysis on other specific biomarkers. The results from the meta-analysis suggest that miR-21 could have a good diagnostic role for colorectal cancer with moderate sensitivity and good specificity; however, we had a small sample of studies that met our inclusion criteria for the meta-analysis. 

MicroRNAs could function as either a tumour suppressors by inhibiting oncogenesis via down-regulation of proteins with oncogenic qualities or promoters by down-regulating tumour suppressors or other genes involved in cell differentiation in several cancers [8]. Another possible role of miRNA is indirect, through the promotion of a pro-carcinogenic cellular environment, such as low-grade inflammation and/or insulin resistance settings.

In particular, the role of miR-21 in carcinogenesis has been extensively investigated in the literature. It is defined as one of the “oncomiRs”, the cancer-promoting miRNAs, and has been documented to target several tumour suppressor genes (such as PTEN, RHOB, PDCD4, TIMP3) and play a role in signalling pathways (RAS/MEK/ERK, PTEN/PI3K/AKT, and Wnt/β-catenin). In colorectal cancer, miR-21 can down-regulate transforming growth factor β receptor 2, inducing stemness, stimulating invasion, and stimulating metastasis by suppressing PDCD4 [59,60,61].

Anyway, studies on the diagnostic role of miRNAs in cancer could only demonstrate an association between the biomarker and colon cancer diagnosis, not a causal role of miRNA in cancer promotion or growth. 

In this systematic review, miR-21 was the most validated miRNA in different studies, confirming a possible role as an early-stage biomarker, stable in different sample sources, and always up-regulated in cancer cases. Further elucidation of the associations between miR-21 and colon carcinogenesis is necessary to understand, in depth, the mechanisms of cancer development and hypothesize therapeutic targets for use in the clinic.

Beyond their role as diagnostic biomarkers, the meet-in-the-middle paradigm suggests that microRNAs could play a role as intermediate biomarkers among environmental exposures and cancer. In fact, miRNA has been associated, in previous studies, with cancer-predisposing conditions, such as obesity, hyperinsulinemia, and inflammation. Identifying the miRNAs that are associated with both CRC and other cancer hallmarks could be useful to disentangle the role of involved factors and to hypothesize a biological pathway from exposure to disease [62].

## 5. Conclusions

This systematic review identifies numerous miRNAs that are associated with CRC; however, little consistency was found between the forty included studies. Only a few miRNAs were reported to be validated in more than one study and no single stand-alone miRNA could yet be defined as an ideal biomarker for the detection of CRC.

Similar to the systematic review, the meta-analysis suggests that more high-quality studies are needed to evaluate the miRNAs associated with colorectal cancer. Further studies on the mechanisms by which miRNAs are biologically embedded in cancer initiation and promotion may help to better understand cancer pathways to develop more effective prevention strategies and therapy approaches.

## Figures and Tables

**Figure 1 biomedicines-10-02224-f001:**
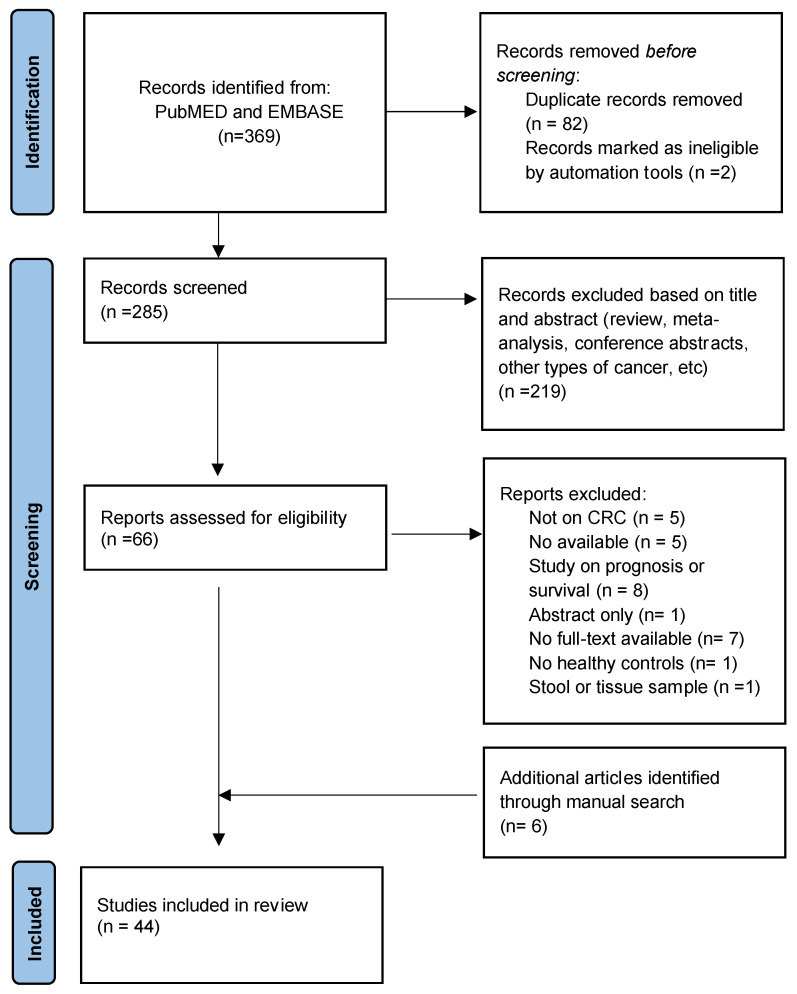
Flowchart of the systematic literature search according to PRISMA guidelines.

**Figure 2 biomedicines-10-02224-f002:**
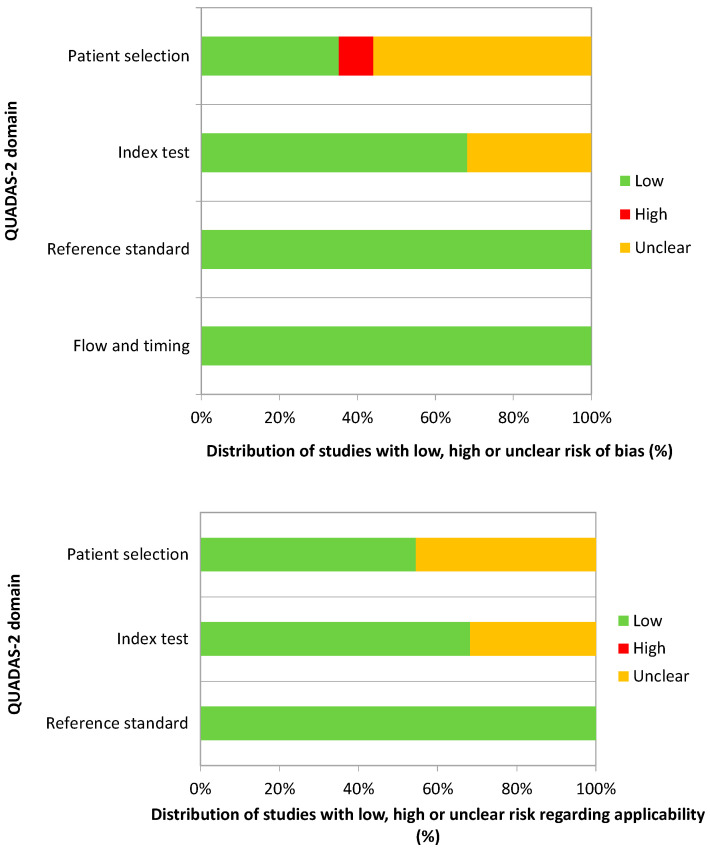
Quality assessment with the QUADAS-2 tool.

**Figure 3 biomedicines-10-02224-f003:**
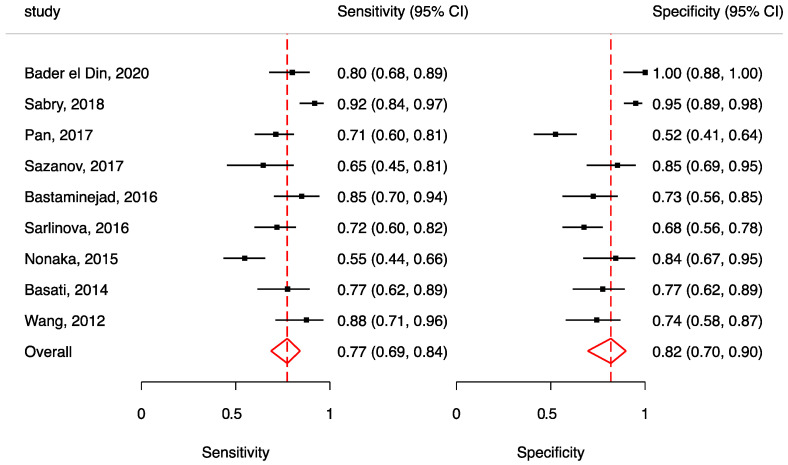
Forest plot of included studies assessing the sensitivity and specificity of miR-21 in CRC diagnosis (squares show sensitivity and specificity, respectively; red diamonds show pooled effect; error bars represent 95% confidence interval) Bader El Din, 2020 [19], Sabry, 2018 [21], Pan, 2017 [36], Sazanov, 2016 [38], Bastaminejad, 2017 [37], Sarlinova, 2016 [42], Nonaka, 2015 [45], Basati, 2014 [52], Wang, 2012 [55].

**Figure 4 biomedicines-10-02224-f004:**
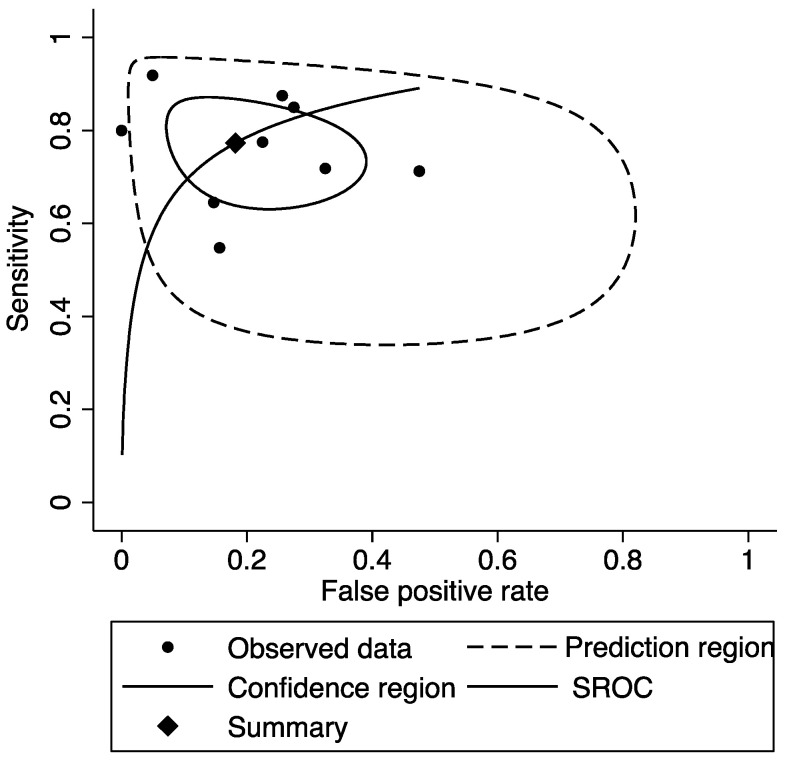
Summary receiver operating characteristic curve (SROC) for miR-21 in CRC diagnosis.

## Data Availability

Not applicable.

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
