# Peer review of "Investigating the Role of Circulating miRNAs as Biomarkers in Colorectal Cancer: An Epidemiological Systematic Review"

_biomedicines, 2022, doi:10.3390/biomedicines10092224_

Round 1

Reviewer 1 Report

This is a well-written review employing a systematic review to identify the role of a few miRNAs in CRC initiation and diagnosis. I recommend publishing this review after some easy fixes as below: 

1. Please clarify why PubMed and EMBASE are the data sources to retrieve targeted publications. and were the keywords used in combination, or respectively? 

2. Please clarify why the title and abstract screening but no full-text review was conducted in the second-round screening. 

3. Please graph the flow diagram of the selection process in a good shape. The modes in the figure 1 diagram are completely off and not in a good shape. 

4. Please include the species of the biological samples (plasma and serum). 

5. Please point out the mechanisms of miR-21 initiating CRC and outline some clinical implications of miR-21 in CRC early diagnosis. 

Author Response

1) Please clarify why PubMed and EMBASE are the data sources to retrieve targeted publications. and were the keywords used in combination, or respectively?

Thank for your comment. The coverage of bibliographic databases varies considerably due to differences in their scope and content. In the field of molecular epidemiology PubMed plus Embase give an optimal coverage of literature, while adding other databases (such as google scholar, Web of Science, etc) give a negligible advantage [Frandsen TB. J Clin Epidemiol, 2021 May;133:24-31.].

We have added in the supplementary material the full search strategy for the systematic search. All the keywords were used in combination.

2) Please clarify why the title and abstract screening but no full-text review was conducted in the second-round screening.

As highlighted in the text and in figure 1 we have conducted both abstract and full-text screening.

3) Please graph the flow diagram of the selection process in a good shape. The modes in the figure 1 diagram are completely off and not in a good shape.

We have amended figure 1.

4) Please include the species of the biological samples (plasma and serum).

We included the Biological speciment type in the table1

5) Please point out the mechanisms of miR-21 initiating CRC and outline some clinical implications of miR-21 in CRC early diagnosis.

We have added in the discussion a section to point out what has been requested.

Reviewer 2 Report

In this review article Dansero et. al., reviewed the publicly available database for microRNA biomarkers for CRC. This article will be helpful for the researcher and clinician working in similar areas of research. Alternatively, this study may be useful for the clinical management of CRC patients. This is a well-designed and well-written article. 

Author Response

Thank you very much for your suggestion. We have amended figure 1.

Round 2

Reviewer 2 Report

None